# Parental Autonomy Support and Mental Health among Chinese Adolescents and Emerging Adults: The Mediating Role of Self-Esteem

**DOI:** 10.3390/ijerph192114029

**Published:** 2022-10-28

**Authors:** Chunhua Ma, Yongfeng Ma, Youpeng Wang

**Affiliations:** 1College of Educational Science and Technology, Northwest Minzu University, Lanzhou 730030, China; 2Department of Petrochemical Engineering, Lanzhou Petrochemical University of Vocational Technology, Lanzhou 730060, China

**Keywords:** mental health, parental autonomy support, self-esteem, adolescents, emerging adults

## Abstract

Guided by the dual-factor model and self-determination theory, this study explored the relationship between parental autonomy support and mental health (i.e., life satisfaction and emotional problems) in adolescents and emerging adults, with a focus on the mediating role of self-esteem. We conducted two studies among independent samples in China, including 1617 adolescents aged 10 to 17 years (*M*age =12.79, *SD* = 1.63; 50.7% girls; Study 1) and 1274 emerging adults aged 17 to 26 years (*M*age = 20.31, *SD* = 1.63; 56.6% women; Study 2). All participants completed a set of self-reported questionnaires. The results of both studies validated our hypothesis; specifically, parental autonomy support was positively associated with life satisfaction, but negatively associated with emotional problems (emotional symptoms in Study 1 and depressive symptoms in Study 2). Meanwhile, self-esteem partially mediated the positive relationship between parental autonomy support and life satisfaction (*R*^2^ = 0.33 in Study 1; *R*^2^ = 0.38 in Study 2), and partially mediated the negative relationship between parental autonomy support and emotional problems (*R*^2^ = 0.16 in Study 1; *R*^2^ = 0.42 in Study 2). In summary, this suggests that the common antecedents of positive and negative indicators of mental health addressed in this study are prevalent in adolescents and emerging adults. These findings have important implications for preventive and interventional efforts aimed at mental health problems in both demographics.

## 1. Introduction

Adolescence and emerging adulthood are important psychosocial development stages [1,2]. In both periods, numerous pressures and challenges emerge because of a range of physical, psychological, and social changes, thus increasing the risk of mental health problems [3,4]. This is a widespread concern, as such problems are prevalent in Eastern and Western societies [5,6,7]. Looking at the conditions in China, a recent study reported a 17.6% rate of mental health problems in a sample exceeding 70,000 adolescents, which is a marked increase from 30 years prior [8], while a similar investigation found a rate ranging from 12.6% to 17.7% in a sample of emerging adults [9]. Given their impacts on multiple functional domains in both groups, mental health problems can impair physical health, hinder cognitive and academic achievement, damage social relationships, and even lead to suicide [10,11,12,13]. These deleterious issues highlight the need for continued research on the correlates of mental health in adolescents and emerging adults.

In the more traditional framework, the presence of mental health refers to the absence of psychopathology and related functional impairments [14,15]. However, this relatively narrow understanding is broadening with the rise of positive psychology. For example, the dual-factor model (DFM) posits that mental health is not limited to a single dimension of illness symptoms (e.g., emotional problems), but also encompasses positive wellness factors (e.g., life satisfaction) [15,16]. While many recent empirical studies have examined mental health using positive and negative indicators, there is still a lack of evidence on their common antecedents. According to self-determination theory (SDT) [17,18], autonomy is an important condition for optimal psychosocial adaptation. In this context, the satisfaction derived from autonomy can also improve an individual’s awareness of self-worth [19]. With these factors in mind, this study adopted DFM and SDT to more comprehensively investigate the correlates of mental health. Specifically, we examined the association between parental autonomy support and mental health (i.e., life satisfaction and emotional problems) among samples of adolescents and emerging adults in China, with a focus on the potential mediating role of self-esteem.

### 1.1. Parental Autonomy Support and Mental Health

According to SDT, autonomy is one of three basic human needs, along with competence and relatedness; it refers to the ability to complete individual actions through one’s own will, and thus reflects the sense of volition [17,18]. Autonomy is an important resource in the domain of individual health, which can stimulate autonomous motivation and enhance adaptive behavior and mental health [20,21]. Satisfaction with autonomy is derived through support from important others, such as parents; here, parental autonomy support arises when parents can respect the wishes of their children, encourage them to express their ideas, and provide them with opportunities to make independent choices [20,22]. Research has shown that parental autonomy support plays an important role in children’s mental health. For example, individuals who perceive the provision of parental autonomy support have higher psychological well-being, better life satisfaction [23,24], and fewer internalizing problems [25,26].

Although previous empirical studies have verified the role of parental autonomy support in individual psychosocial development, some issues remain unclear. First, most research has focused on conditions during adolescence, perhaps because of the importance of this period in autonomy development [2,27]. As adolescents require more support and guidance, parents should alter their protective parenting styles to accommodate new characteristics [27]. From this perspective, parental autonomy support has special significance. Emerging adults are also in a period of identity exploration, wherein autonomy can lead to a sense of choice in development and other aspects of regulation [1,2]. In this context, parental autonomy support can facilitate better engagement in identity choices. This emphasizes the need for more research among emerging adults, for whom parents still play integral roles in multiple functional domains and continue to be seen as critical support sources in many important matters [2,28]. Even when emerging adults have moved away from their parents, the parental influence remains a significant predictor of psychosocial developmental outcomes [28]. Given this evidence, this study was divided into two components to focus on adolescents and emerging adults.

Second, most relevant studies have focused on samples in Western cultures, which emphasize autonomy and self-expression [29]. By contrast, Chinese culture is influenced by Confucianism, which emphasizes societal harmony and interdependence [30]. These differences in cultural values are partially transmitted through parenting [27,31]. In China, parent–child relationships are, therefore, centered around parental control and filial piety [32,33]. However, some recent studies have shown that Chinese parenting styles emphasize both autonomy support and high involvement and that parental autonomy support also plays an important role in the Chinese cultural context [34,35]. Given the above evidence, this study further explored how parental autonomy support affected mental health in China.

### 1.2. Self-Esteem as a Mediator

Self-esteem involves subjective evaluations of one’s personal worth, and thus reflects self-acceptance and positive attitudes toward the self [36]. Its development is important during all life stages, especially in adolescence and emerging adulthood. Once adolescents gain self-awareness, autonomy, and a sense of control, they are eager to receive respect and acceptance from others; while self-esteem rapidly develops under these conditions [37,38,39], it is unstable, variable, and easily affected by external factors [37,38]. Subsequently, social roles tend to change for emerging adults, such that their personality traits often develop toward maturity [39,40,41]. As a personality trait, self-esteem also exhibits relatively strong normative growth during emerging adulthood [39,40], which helps individuals address various problems [40,41]. As self-esteem is characterized by dynamic changes that occur from adolescence to adulthood, this study investigated its age-related roles in the relationship between parental autonomy support and mental health.

Given the large body of evidence showing that self-esteem is related to both parental autonomy support and mental health, it may be important for understanding how the two factors are linked. First, the family is an important system for cultivating self-esteem, which is encouraged under parenting styles that lend autonomy support [42]. According to SDT, this provision not only meets the need for autonomy in children but also facilitates identity development and the realization of self-worth [20,43]. Second, self-esteem is associated with many aspects of individual mental health. For example, it promotes positive psychological functions such as well-being and life satisfaction [40,44] while reducing negative psychological disorders such as depression and anxiety [45,46].

While previous studies have demonstrated the existence of relations between self-esteem, parental autonomy support, and mental health, it is unclear whether self-esteem plays a mediating role. However, self-esteem is known to play mediating roles in related domains. For example, Peng et al. [47] found that self-esteem played a mediating role in the relation between the autonomy support of important others and problem behaviors. To explain this, the researchers argued that satisfaction with autonomy facilitated self-integration and self-evaluation, thus aiding the avoidance of problem behaviors. Other studies have reported that self-esteem mediates the relation between parenting style and subjective well-being [48,49]. In this context, feelings of acceptance and respect can promote self-esteem, which is increasingly conducive to life satisfaction. Based on this evidence, we posit that self-esteem plays a mediating role in the relationship between parental autonomy support and mental health.

### 1.3. Overview of the Two Study Components

Guided by DFM and SDT, we established a structural equation model for parental autonomy support, self-esteem, and mental health (i.e., life satisfaction and emotional problems). This allowed us to test the relationship between autonomy support and mental health, with a focus on the potential mediating role of self-esteem (see Figure 1). The purpose of this study is to demonstrate that our findings can be extended to different age groups using two independent samples. The first study’s objective was to explore the direct and indirect relations of parental autonomy support and self-esteem with adolescent mental health. The second study aimed to replicate the study associations obtained above and extend the first study to demonstrate whether these under-investigated relationships exist in other age groups, such as emerging adults. If converging evidence can be gathered across both independent studies, the robustness of study associations can be justified and the commonality of these relationships in different age groups can be validated. Based on relevant theories and a literature review, we posited the following hypotheses:H1: Parental autonomy support is positively associated with life satisfaction and negatively associated with emotional problems (i.e., emotional symptoms in Study 1 and depressive symptoms in Study 2);H2: Self-esteem mediates the relation between parental autonomy support and mental health; specifically, autonomy support is positively associated with self-esteem, which is then positively associated with life satisfaction (H2a) but negatively associated with emotional problems (H2b).

**Figure 1 ijerph-19-14029-f001:**
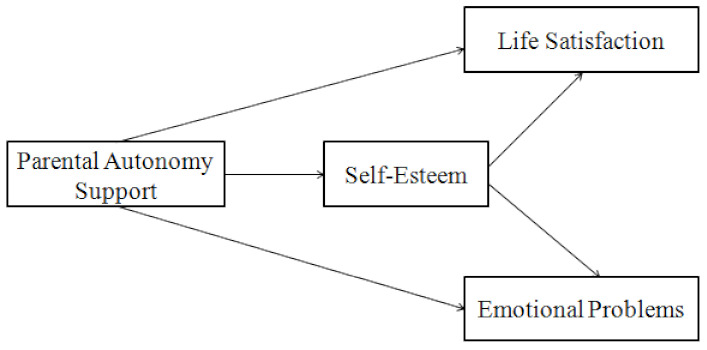
Hypothesized model.

## 2. Study 1

In Study 1, we explored our hypotheses among a sample of adolescents; that is, we tested whether parental autonomy support was related to their mental health and whether self-esteem mediated this relationship.

### 2.1. Study 1 Methods

#### Participants and Procedures

After obtaining consent from the research ethics committee at the first author’s university, we contacted the leaders of two primary and secondary schools each in the city of Linxia, which is located in northwestern China, Gansu Province. The survey participants included students in grades four through eight (those in grade nine did not participate because of busy academic schedules). We first informed the participants that their data and results would only be used for scientific research. Moreover, we fully followed the principle of confidentiality, such that all participation was both voluntary and anonymous. After obtaining consent from all participants, we conducted the survey in a group format. The researcher’s assistant uniformly distributed the questionnaires during class. These were returned after completion, which took approximately 20 min.

Study 1 initially included 1635 participants. After removing cases with missing crucial information (e.g., gender and age), we obtained valid responses from 1617 participants aged 10 to 17 years (*M_age_* = 12.79, *SD* = 1.63; 50.7% girls) and with moderate family socioeconomic status (SES; *M_SES_* = 4.47, *SD* = 2.07).

### 2.2. Study 1 Measures

#### 2.2.1. Parental Autonomy Support

We measured parental autonomy support via the parental autonomy support questionnaire developed by Wang et al. [50]. Participants rated each of the eight total items (e.g., “My parents allow me to plan what I want to do”) using a 5-point Likert scale ranging from 1 (completely disagree) to 5 (completely agree). Average scores were then calculated, such that higher scores represented higher parental autonomy support. Previous studies have shown this scale to have good psychometric properties for use among Chinese adolescents and adults [50,51]. In this study, Cronbach’s alpha was 0.94.

#### 2.2.2. Self-Esteem

We measured self-esteem using the Self-Esteem Scale developed by Rosenberg [36]. Participants rated each of the 10 total items (e.g., “I take a positive attitude toward myself”) using a 4-point Likert scale ranging from 1 (strongly disagree) to 4 (strongly agree). Average scores were then calculated, such that higher scores represented higher self-esteem. Previous research has shown that the scale is reliable for use among Chinese samples [52]. Cronbach’s alpha in this study was 0.74.

#### 2.2.3. Life Satisfaction

We measured life satisfaction using the Satisfaction with Life Scale (SWLS) from Diener et al. [53]. Participants rated each of the five items (e.g., “In most ways, my life is close to my ideal”) using a 5-point Likert scale ranging from 1 (totally disagree) to 5 (totally agree). Average scores were then calculated, such that higher scores represented higher life satisfaction. Previous research has shown the scale to have good internal consistency for use among Chinese samples [54]. Cronbach’s alpha in this study was 0.76.

#### 2.2.4. Emotional Symptoms

We measured emotional symptoms via the emotional symptoms subscale of the Strengths and Difficulties Questionnaire (SDQ) from Goodman [55]. Participants rated each of the five total items using a 3-point Likert scale, with options ranging from 0 (not true) to 2 (certainly true). Total scores were then calculated, such that higher scores represented higher levels of emotional symptoms. The scale was previously validated for use in China [56]. Cronbach’s alpha in this study was 0.71.

#### 2.2.5. Covariates

We also investigated demographic variables such as gender, age, and family socioeconomic status (SES). There were several reasons to establish these factors as covariates. According to previous research, girls are more prone to emotional problems than boys [57], and also tend to have relatively higher well-being [58]. Meanwhile, age is related to emotional disorders and subjective well-being [4,58], while family SES is associated with adolescent mental health [59].

We used the FAS [60] to measure family SES. Previous research has used the FAS as a proxy for family SES because it is easier for adolescents to understand the items [52]. In this study, participants rated the four total items (e.g., “Do you have your own bedroom?”) using a scale ranging from 0 (none) to 3 (more than two). Total scores were then analyzed, such that higher scores represented higher family wealth.

### 2.3. Study 1 Data Analytics Plan

We used IMB SPSS 25.0 and the Amos 17.0 statistical software for data analysis. First, we performed descriptive statistics and a Pearson correlation analysis for all study variables. Second, we established a structural equation model to analyze the effects of autonomy support on life satisfaction and emotional symptoms. Third, we established a mediation model for self-esteem to analyze its potential mediating effects, both in the relation between autonomy support and life satisfaction and in the relation between autonomy support and emotional symptoms. The mediation model was performed using the maximum likelihood estimation method, with evaluations for model fit indices. Since *χ^2^* is easily affected by the sample size, we also calculated absolute fit indices. In general, absolute fit indices (e.g., CFI and TLI) should be higher than 0.90, while RMSEA should be lower than 0.08 [61]. Finally, we estimated the mediating effect of self-esteem using the bootstrap method with 5000 samples. Mediation effects were considered significant in cases where 95% confidence intervals (CIs) did not contain zero.

### 2.4. Study 1 Results

#### 2.4.1. Descriptive Statistics and Correlations

Table 1 shows the descriptive statistics and Pearson’s correlation for the study variables. We found that parental autonomy support was positively correlated with self-esteem and life satisfaction, while parental autonomy support, self-esteem, and life satisfaction were negatively correlated with emotional symptoms. Furthermore, age was negatively correlated with parental autonomy support, self-esteem, and life satisfaction, but positively correlated with emotional symptoms. Compared with boys, girls had higher emotional symptoms and lower autonomy support, self-esteem, and life satisfaction. SES was positively correlated with parental autonomy support, self-esteem, and life satisfaction, but negatively correlated with emotional symptoms. We also measured skewness and kurtosis for each variable, with values ranging from −1 to 1, thus finding that the data did not violate the normality assumption [62].

In addition, we tested measurement invariance by gender. To this end, we compared the changes in CFI and RMSEA between unconstrained and constrained models. Once the unconstrained model provides a good fit to confirm the configural invariance, the metric and scalar invariances are subsequently tested. If the changes of CFI do not exceed 0.01, and the changes of RMSEA are less than 0.015, the measurement invariance across genders is assumed to be achieved. The detailed results are presented in Appendix A.

#### 2.4.2. Parental Autonomy Support and Mental Health in Adolescents

After controlling for gender, age, and SES, we established a structural equation model to analyze the relationship between autonomy support and mental health. The results showed that this model was acceptable (*χ^2^*/*df* = 3.54, NFI = 0.99, TLI = 0.96, CFI = 0.99, RMSEA = 0.04). As for the total effect, autonomy support was positively associated with life satisfaction (*β* = 0.45, *p* < 0.01), but negatively associated with emotional symptoms (*β* = −0.23, *p* < 0.001). This finding verified H1.

#### 2.4.3. Mediation Analyses

Based on our hypothesis, we established a mediation model with self-esteem set as the mediating variable. The model fit well (*χ^2^*/*df* = 7.78, NFI = 0.97, TLI = 0.90, CFI = 0.98, RMSEA = 0.05). Autonomy support was positively associated with life satisfaction (*β* = 0.37, *p* < 0.001), but negatively associated with emotional symptoms (*β* = −0.14, *p* < 0.001); here, the direct effects were significant. Moreover, parental autonomy support was positively associated with self-esteem (*β* = 0.34, *p* < 0.001), which was then positively associated with life satisfaction (*β* = 0.23, *p* < 0.001), but negatively associated with emotional symptoms (*β* = −0.28, *p* < 0.001).

We also tested the mediating effects of self-esteem using the bootstrap method with 5000 samples. The indirect effect between autonomy support and life satisfaction was 0.08 (95% *CI* = 0.06, 0.10), while the indirect effect between autonomy support and emotional symptoms was −0.10 (95% *CI* = −0.12, −0.02). As the 95% CIs did not contain zero, the indirect effects were significant. Thus, self-esteem partially mediated the positive relationship between parental autonomy support and life satisfaction (*R*^2^ = 0.33), and partially mediated the negative relationship between parental autonomy support and emotional problems (*R*^2^ = 0.16). This finding verified H2 (Table 2 and Figure 2).

### 2.5. Brief Discussion on Study 1

According to DFM, we divided mental health into positive (life satisfaction) and negative (emotional symptoms) indicators. We explored the common antecedents of these factors with a large sample of adolescents, thus validating our two study hypotheses; that is, parental autonomy support was positively associated with life satisfaction, but negatively associated with emotional symptoms, while self-esteem partially mediated both relations.

## 3. Study 2

In Study 2, we replicated Study 1 among an independent sample of emerging adults, thus allowing us to compare the results between age groups. Here, we collected data via online questionnaires. To more comprehensively control for family SES, we also collected information on family income, parental education background, and parental occupation [63,64]. Since depression is a more prevalent emotional disorder in emerging adults [7,13], we used this factor to represent the negative indicator (emotional problems) of mental health.

### 3.1. Study 2 Methods

#### Participants and Procedures

Before commencement, we obtained approval from the Research Ethics Committee at the first author’s university. We then contacted professors at two universities in the city of Lanzhou, which is located in northwestern China, Gansu Province. In turn, these professors contacted university students to inquire about their willingness to participate in this study. We used the same confidentiality and consent processes described for Study 1. Finally, we distributed online questionnaires via WeChat between classes. In each case, answering times ranged from 10 to 15 min.

The initial sample included 1279 emerging adults. Since the online questionnaires required full completion before submission, all were collected. After removing five duplicate cases, the final effective sample size was 1274 (*M_age_* = 20.31, *SD* = 1.63, 56.6% women, age range of 17 to 26 years). As for the parents of participants, 49.2% of their fathers and 43.8% of their mothers had completed high school education or higher, with moderate family monthly income (*M* = 4.03, *SD* = 1.67; a range of 1 to 7).

### 3.2. Study 2 Measures

#### 3.2.1. Parental Autonomy Support

We assessed parental autonomy support using the parental autonomy support questionnaire [50] (see Study 1 for details). In Study 2, Cronbach’s alpha was 0.96.

#### 3.2.2. Self-Esteem

We assessed self-esteem using the Self-Esteem Scale developed by Rosenberg [36] (see Study 1 for details). In Study 2, Cronbach’s alpha was 0.92.

#### 3.2.3. Life Satisfaction

We assessed life satisfaction using the SWLS [53] (see Study 1 for details). In Study 2, Cronbach’s alpha was 0.92.

#### 3.2.4. Depressive Symptoms

We assessed depressive symptoms using the Center for Epidemiological Studies (CES-D) Scale [65], which contains 20 items (e.g., “I was bothered by things that usually do not bother me”). Participants rated each item using a 4-point Likert scale ranging from 0 (rarely) to 3 (sometimes). Total scores were then calculated, such that higher scores indicated higher levels of depression. Studies have shown the scale to have good internal consistency for use among Chinese adults [66]. In this study, Cronbach’s alpha was 0.84.

#### 3.2.5. Covariates

We also investigated demographic variables such as gender, age, and SES. According to previous research, these variables are related to emotional problems and life satisfaction in adults [4,67,68]. As such, we controlled for each in our analyses.

Family SES included parental education background (four categories), occupation (seven categories), and family monthly income (seven categories). We then standardized the three scores and calculated the total scores to represent family SES [63,64].

### 3.3. Study 2 Data Analytics Plan

In Study 2, we used the same data analytics plan described in Study 1.

### 3.4. Study 2 Results

#### 3.4.1. Descriptive Statistics and Correlations

Table 3 shows that parental autonomy support was positively correlated with self-esteem and life satisfaction, while parental autonomy support, self-esteem, and life satisfaction were negatively correlated with depressive symptoms. Furthermore, age was negatively correlated with life satisfaction. Compared with men, women had higher parental autonomy support, lower life satisfaction, and lower depressive symptoms. SES was positively correlated with self-esteem. Moreover, the skewness and kurtosis values were in the range of −1 to 1.

We also investigated measurement invariance by gender in this study, and the detailed results are presented in Appendix A.

#### 3.4.2. Parental Autonomy Support and Mental Health in Emerging Adults

After controlling for gender, age, and SES, we established a structural equation model. The results showed that this model fit well (*χ^2^*/*df* = 3.08, NFI = 0.98, TLI = 0.96, CFI = 0.99, RMSEA = 0.04). As for the total effect, autonomy support was positively associated with life satisfaction (*β* = 0.52, *p* < 0.001), but negatively associated with depressive symptoms (*β* = −0.50, *p* < 0.001). This finding verified H1.

#### 3.4.3. Mediation Analyses

Based on our hypothesis, we established a mediation model with self-esteem set as the mediating variable. The model fit well (*χ^2^*/*df* = 3.83, NFI = 0.99, TLI = 0.97, CFI = 0.99, RMSEA = 0.05). Parental autonomy support was positively associated with life satisfaction (*β* = 0.28, *p* < 0.001), but negatively associated with depressive symptoms (*β* = −0.12, *p* < 0.01); here, the direct effects were significant. Autonomy support was positively associated with self-esteem (*β* = 0.69, *p* < 0.001), which was then positively associated with life satisfaction (*β* = 0.35, *p* < 0.001), but negatively associated with depressive symptoms (*β* = −0.55, *p* < 0.001).

We tested the mediating effect of self-esteem using the bootstrap method with 5000 samples. The indirect effect between autonomy support and life satisfaction was 0.24 (95% *CI* = 0.20, 0.30), while the indirect effect between autonomy support and depressive symptoms was −0.38 (95% *CI* = −0.43, −0.33). As the 95% CIs did not contain zero, the indirect effects were significant. Thus, self-esteem partially mediated the positive relationship between parental autonomy support and life satisfaction (*R*^2^ = 0.38), and partially mediated the negative relationship between parental autonomy support and depressive symptoms (*R*^2^ = 0.42). This finding verified H2 (Table 4 and Figure 3).

### 3.5. Brief Discussion on Study 2

According to DFM, we divided mental health into positive (life satisfaction) and negative (depressive symptoms) indicators. Thus, we replicated Study 1 among a large sample of emerging adults. Our findings confirmed our hypotheses. These findings underscore the importance of both parental autonomy support and self-esteem for mental health in adolescents and emerging adults.

## 4. General Discussion

Mental health problems are prevalent in adolescents and emerging adults, with many negative effects on both groups. Although the literature offers important findings in this context, there is limited evidence on the common antecedents of positive and negative indicators of mental health. In this study, we adopted both DFM and SDT to separately investigate the association between parental autonomy support and mental health among two independent samples of adolescents and emerging adults in China, with a focus on the mediating role of self-esteem. The results of both studies validated our hypotheses. Specifically, parental autonomy support was positively associated with life satisfaction but negatively associated with emotional problems, while self-esteem partially mediated both relations.

We first explored the relationship between parental autonomy support and mental health, wherein we found support for H1 in both studies. This is consistent with DFM, SDT, and previous findings [16,17,18]. Under SDT, a possible explanation is that parental autonomy support helps children feel understood and respected, which then satisfies their need for autonomy and increases life satisfaction [17,18,69]. Simultaneously, individuals who receive autonomy support have better self-regulation skills and show more progress toward goal achievement, which reduces the risk of negative emotions [20,21]. Further, our results confirmed that parental autonomy support was important for mental health, both in adolescents and emerging adults. A possible explanation is that adolescence and emerging adulthood are two critical stages for autonomy development. For adolescents, autonomy marks the beginnings of personal independence, which entails more opportunities to decide one’s own behavior without parental influence, and parental autonomy support plays a fundamental role during this stage [2]. Meanwhile, emerging adults tend to have more independence from their childhood families, but there is still an urgent need to further develop autonomy and independence, in which case parental autonomy support can still provide help [70]. Finally, our results are consistent with previous findings in Eastern and Western cultural contexts [34,35,69]. In the Chinese cultural context, we provide additional evidence that autonomy support is highly significant in the development of positive functioning, thus demonstrating the applicability of SDT.

Second, we explored the mediating role of self-esteem. The results of both studies supported H2. Our finding of a positive association between autonomy support and self-esteem is consistent with both SDT and previous findings [19,42]. A possible reason for this is that self-esteem, as an individual’s self-awareness and evaluation, often depends on the evaluation of important others. In this context, individuals who are respected and accepted by important others have more positive self-awareness and self-evaluations, but incur negative impacts in cases of rejection or criticism [71,72]. Another possible explanation is that self-esteem exhibits dynamic developmental changes from adolescence to emerging adulthood, but parental autonomy support (a prerequisite for positive psychological functioning) still plays an important role in its formation during both stages. For adolescents, self-esteem may become more stable and develop once autonomy is perceived [37,39]. For emerging adults, parental autonomy support may motivate full engagement in opportunities to develop one’s personality, thereby contributing to self-esteem [39,73].

Third, self-esteem can promote mental health, as it is both positively associated with life satisfaction and negatively associated with emotional problems. A possible reason for this is that individuals with high self-esteem have more positive self-awareness and view current and future life with a positive attitude. In turn, they are more satisfied with life and less troubled by emotional problems [74]. By contrast, individuals with low self-esteem may repeatedly think about their negative aspects and thus engage in negative self-evaluation (e.g., “I am a useless person”, “I am worthless”); this increases the potential for loneliness and depression, which can lead to emotional disorders and negative attitudes about life [45,46,74].

Finally, family SES may influence the manifestation of parental autonomy support, which may further impact the direct and indirect correlations with individuals’ mental health under investigation. Family SES has been shown to be positively related to parental autonomy support, particularly in adolescents, as evidenced by the correlation analyses in Study 1. This is consistent with previous research indicating that SES can influence parenting practices [75,76,77]. One possible explanation is that parents from higher SES families tend to use positive practices, such as autonomy support, more than those from lower SES families. By gaining psychological benefits from autonomy support, individuals are less likely to develop mental health difficulties. In sharp contrast, parents from low SES families often grapple with livelihood and encounter various life challenges due to restrained family economic situations [76,77]. Parents in this context may have limited time and energy in supervising their children using autonomy-supportive strategies because they may have difficulties with handling their own psychological distress [75]. Consequently, children raised in low SES families are more likely to suffer from mental health challenges due to a lack of autonomy support. In the current study, participants came from middle-class families, and thus the beneficial role of parental autonomy support in their mental health might be more pronounced.

### Limitations and Implications

While this study made important findings on the correlates of mental health among adolescents and emerging adults, there were also some limitations. First, we collected cross-sectional data, which limits inferences on causality. Future longitudinal studies are therefore needed. Second, we met the necessary psychometric requirements, but self-reported questionnaires may entail common methodological biases and social approval. Thus, multi-method studies can increase reliability. Third, our participants were solely recruited from schools and universities in northwestern China, which limits generalizability to the national population. Future studies should therefore explore different regions. Finally, in this study, we statistically controlled for the levels of family SES by regressing these values on self-esteem and outcomes. Given that family SES may directly influence the manifestation of parental autonomy support, future studies should clarify the impact of family SES in the tested relationships by recruiting a more diverse sample with distinct socioeconomic characteristics.

This study contains important practical implications that should be taken into account for mental health interventions among adolescents and emerging adults. First, educators or school-based psychologists should organize structured meetings with parents, either on-site or online, to introduce the concept of autonomy support and highlight the benefits of such interpersonal interaction styles for their children’s mental health. This introduction module is important for better conceptual understanding at the initial stage. During the following sections of these meetings, educators or school psychologists may provide specific guidance and strategies to parents regarding how to internalize these autonomy-oriented practices when interacting with their children. For instance, parents should welcome their children’s thoughts and attempt to incorporate their children’s perspectives when making decisions for them. Although parents should appropriately adjust their children’s irrational claims, they are recommended to use non-evaluative, non-controlling, flexible language; parents should also provide possible solutions and present solid rationales when supervising their children on those decisions.

Second, educators or school psychologists should pay attention to the bridging role of self-esteem, particularly considering that, during adolescence and emerging adulthood, positive self-regard formation is crucial to student healthy development. Some school-based activities should be organized in this regard. For example, educators or school psychologists should let students write down a list of moments when they feel well and a few things of which they feel proud. It is essential to reinforce these positive moments and experiences, stimulating positive perspectives on themselves.

Finally, during these initiatives, educators or school psychologists should regularly assess students’ perceptions of parental autonomy support and self-esteem. For those who consistently exhibit low parental autonomy support and self-esteem, educators or school psychologists may consider organizing individual psychological counseling sessions. In these scenarios, students and parents are encouraged to come together to talk with school psychologists. Professionals could better help them identify possible difficulties with these implementations and ultimately find ways to handle these challenges that may affect how parents create autonomy-supportive styles and students feel about themselves.

## 5. Conclusions

The present research underscores the importance of both parental autonomy support and self-esteem in mental health among adolescents and emerging adults. In practice, efforts must be made to change parenting styles and improve individual self-esteem, especially in the context of interventions and other preventive efforts aimed at mental health problems.

## Figures and Tables

**Figure 2 ijerph-19-14029-f002:**
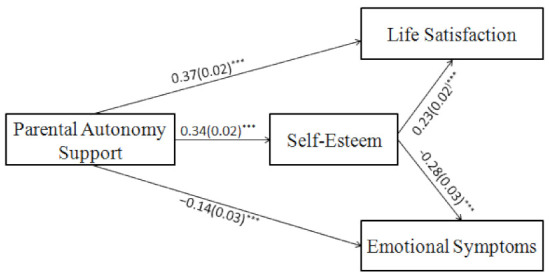
Mediation model for Study 1. *Note. N* = 1617. Standardized path coefficients and standard errors (inside the brackets); *** *p* < 0.001.

**Figure 3 ijerph-19-14029-f003:**
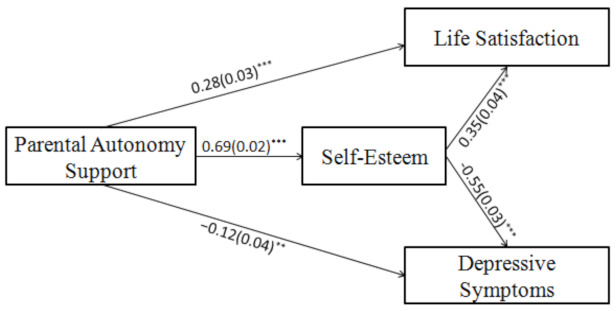
Mediation model for Study 2. *Note. N* = 1274. Standardized path coefficients and standard errors (inside the brackets); ** *p* < 0.01, *** *p* < 0.001.

**Table 1 ijerph-19-14029-t001:** Descriptive statistics and bivariate correlations between variables in Study 1.

Variable	*M*	*SD*	Range	Skewness	Kurtosis	1	2	3	4	5	6	7
1.PAS	3.61	0.87	1–5	−0.71	0.44	-						
2. SE	2.91	0.48	1–4	−0.09	−0.46	0.37 **	-					
3. LS	4.35	1.23	1–7	−0.05	−0.23	0.49 **	0.40 **	-				
4. EP	4.35	2.65	0–10	0.19	−0.81	−0.26 **	−0.35 **	−0.21 **	-			
5. Age	12.79	1.63	10–17	−0.01	−0.99	−0.17 **	−0.15 **	−0.26 **	0.14 **	-		
6. Gender ^a^	-	-	1–2	-	-	−0.05	−0.07 **	−0.05 *	0.14 **	0.02	-	
7. SES	4.47	2.07	0–9	−0.14	−0.59	0.15 **	0.16 **	0.24 **	−0.07 **	−0.30 **	0.004	-

*Note. N* = 1617. ^a^ coded as 1 = boys, 2 = girls. PAS = parental autonomy support; SE = self-esteem; LS = life satisfaction; EP = emotional symptoms; SES = socioeconomic status; * *p* < 0.05, ** *p* < 0.01.

**Table 2 ijerph-19-14029-t002:** Mediation model results for Study 1.

Variable	Estimated	*SE ^a^*	95 % *CI*		*p*
			Lower	Upper	
PAS→LS (total)	0.45 ***	0.02	0.41	0.49	<0.01
PAS→LS (direct)	0.37 ***	0.02	0.33	0.42	<0.001
PAS→SE	0.34 ***	0.02	0.30	0.39	<0.001
SE→LS	0.23 ***	0.02	0.18	0.27	<0.001
PAS→SE→LS (indirect)	0.08 ***	0.01	0.06	0.10	<0.001
PAS→EP (total)	−0.23 ***	0.03	−0.29	−0.19	<0.001
PAS→EP (direct)	−0.14 ***	0.03	−0.19	−0.09	<0.001
PAS→SE	0.34 ***	0.02	0.30	0.39	<0.001
SE→EP	−0.28 **	0.03	−0.33	−0.23	<0.001
PAS→SE→EP (indirect)	−0.10 ***	0.01	−0.12	−0.07	<0.001

*Note. N* = 1617. PAS = parental autonomy support; SE = self-esteem; LS = life satisfaction; EP = emotional symptoms; *SE*
*^a^* = standard error; ** *p* < 0.01, *** *p* < 0.001.

**Table 3 ijerph-19-14029-t003:** Descriptive statistics and bivariate correlations between variables in Study 2.

Variable	*M*	*SD*	Range	Skewness	Kurtosis	1	2	3	4	5	6	7
1.PAS	3.69	0.91	1–5	−0.92	0.06	-						
2. SE	2.92	0.64	1–4	−0.77	0.05	0.69 **	-					
3. LS	3.66	1.35	1–7	0.24	−0.63	0.51 **	0.54 **	-				
4. DS	38.03	10.35	16–80	0.97	1.02	−0.50 **	−0.63 **	−0.30 **	-			
5. Age	20.31	1.63	17–26	0.51	−0.28	−0.01	0.03	−0.08 **	0.02	-		
6. Gender ^a^	-	-	1–2	-	-	0.07 *	0.03	−0.06 *	−0.08 **	−0.04	-	
7. SES	0	1.93	−6.8–8.62	0.54	0.33	0.02	0.04	0.23 **	0.04	−0.06 *	−0.08 **	-

*Note. N* = 1274. ^a^ coded as 1 = men, 2 = women. PAS = parental autonomy support; SE = self-esteem; LS = life satisfaction; DS = depressive symptoms; SES = socioeconomic status * *p* < 0.05, ** *p* < 0.01.

**Table 4 ijerph-19-14029-t004:** Mediation model results for Study 2.

Variable	Estimated	*SE ^a^*	95 % *CI*		*p*
			Lower	Upper	
PAS→LS (total)	0.52 ***	0.02	0.48	0.56	<0.001
PAS→LS (direct)	0.28 ***	0.03	0.21	0.34	<0.001
PAS→SE	0.69 ***	0.02	0.65	0.73	<0.001
SE→LS	0.35 ***	0.04	0.29	0.42	<0.001
PAS→SE→LS (indirect)	0.24 ***	0.03	0.20	0.30	<0.001
PAS→DS (total)	−0.50 ***	0.03	−0.56	−0.44	<0.001
PAS→DS (direct)	−0.12 **	0.04	−0.19	−0.04	0.001
PAS→SE	0.69 ***	0.02	0.65	0.73	<0.001
SE→DS	−0.55 ***	0.03	−0.61	−0.50	<0.001
PAS→SE→DS (indirect)	−0.38 ***	0.03	−0.43	−0.33	<0.001

*Note. N* = 1274. PAS = parental autonomy support; SE = self-esteem; LS = life satisfaction; DS = depressive symptoms; *SE*
*^a^* = standard error; ** *p* < 0.01, *** *p* < 0.001.

## Data Availability

The data presented in this study are available on request from the first author.

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
