# Peer review of "Parental Autonomy Support and Mental Health among Chinese Adolescents and Emerging Adults: The Mediating Role of Self-Esteem"

_ijerph, 2022, doi:10.3390/ijerph192114029_

Round 1

Reviewer 1 Report

This study aimed to test the mediating effect of self-esteem in the relationship among parental autonomy support between mental health specially for life satisfaction, and emotional problem through a structural equation modeling (SEM) in 2 independent samples of adolescents and emerging adults in Chinese. Overall, this is a well-written study and has potential to provide a contribution to the literature if the following comments are addressed.

Major concerns:

1.          There are many statistical methods for mediation analysis and I was wondering why the authors chose SEM approach to address it?

2.          One of the limitations for this manuscript was the authors tested this SEM model separately in two studies and it seems due to inconsistent measures of emotional problem between two studies. Thus, the investigators were not able to test if the SEM model was invariant between two age populations. However, I would like to suggest the investigators test measurement invariance by gender.  

3.          Would expect the authors to clarify the results of their mediation analysis re self-esteem either fully or partially mediated the path between parental autonomy support and mental health and expect to add the variance accounted for self-esteem in the Abstract.     

4.          Would expect to provide some discussion how the findings from this study about the mediating effect of self-esteem to inform the intervention for adolescents and emerging adults in Chinese.

Author Response

Thank you for your feedback and constructive comments. Based on your valuable suggestions, we have now revised the manuscript, as detailed below.  

  1. There are many statistical methods for mediation analysis and I was wondering why the authors chose SEM approach to address it?

RESPONSE: Thank you for this comment. In this study, we chose SEM for performing mediation analysis based on the following considerations. First, this study involved multiple dependent variables (i.e., life satisfaction and emotional problems). In this regard, SEM could simultaneously estimate the direct and indirect relations of parental autonomy support and self-esteem with both life satisfaction and emotional problems, taking the potential co-correlation between life satisfaction and emotional problems into account (Gunzler et al., 2013). This co-correlation is frequently ignored in traditional regression analysis as each dependent variable under this analytical approach should be entered into the model separately. Second, SEM allows researchers to estimate not only the specific path coefficients in the model but also the overall model fit. This approach could provide a holistic picture of the model tested, exhibiting how well the conceptual model fits empirical data collected (Gunzler et al., 2013). Therefore, we proposed that SEM was more appropriate than the traditional analytical approach (e.g., linear regression) to estimate study associations.

  1. One of the limitations for this manuscript was the authors tested this SEM model separately in two studies and it seems due to inconsistent measures of emotional problem between two studies. Thus, the investigators were not able to test if the SEM model was invariant between two age populations. However, I would like to suggest the investigators test measurement invariance by gender.  

RESPONSE: Thank you for pointing out this limitation. The purpose of this study is to demonstrate that our findings can be extended to different age groups using two independent samples. The first study's objective was to explore the direct and indirect relations of parental autonomy support and self-esteem with adolescent mental health. The second study aimed to replicate study associations obtained above and extend the first study to demonstrate whether these under-investigated relationships exist in other age groups, such as emerging adults. If converging evidence can be gathered across both independent studies, the robustness of study associations can be justified and the commonality of these relationships in different age groups can be validated. We carefully revised the “Overview of Studies” section to clarify these perspectives (see p. 3). 

Furthermore, based on your valuable suggestions, we tested measurement invariance by genders (see p. 6 and p.10). The detailed results have been supplemented in Table S1 and S2.

REVISION: The purpose of this study is to demonstrate that our findings can be extended to different age groups using two independent samples. The first study's objective was to explore the direct and indirect relations of parental autonomy support and self-esteem with adolescent mental health. The second study aimed to replicate study associations obtained above and extend the first study to demonstrate whether these under-investigated relationships exist in other age groups, such as emerging adults. If converging evidence can be gathered across both independent studies, the robustness of study associations can be justified and the commonality of these relationships in different age groups can be validated. (see p. 3).

We tested measurement invariance by genders. To this end, we compared the changes in CFI and RMSEA between unconstained and constrained models. Once the unconstrained model provides a good fit to confirm the configural invariance, the metric and scalar invariances are subsequently tested. If the changes of CFI do not exceed 0.01, and the changes of RMSEA are less than 0.015, the measurement invariance across genders is assumed to be achieved (see p. 6 and p.10). The results in details have been supplemented in Table S1 and S2.

  1. Would expect the authors to clarify the results of their mediation analysis re self-esteem either fully or partially mediated the path between parental autonomy support and mental health and expect to add the variance accounted for self-esteem in the Abstract.     

RESPONSE: Thank you for your valuable suggestions. We updated the following details to the Abstract and clarified that self-esteem partially mediated the relationship between parental autonomy support and mental health. (see p. 1).

REVISION: self-esteem partially mediated the positive relation between parental autonomy support and life satisfaction (R2 = 0.33 in Study 1; R2 = 0.29 in Study 2), and partially mediated the negative relation between parental autonomy support and emotional problems (R2 = 0.16 in Study 1; R2 = 0.36 in Study 2).

  1. Would expect to provide some discussion how the findings from this study about the mediating effect of self-esteem to inform the intervention for adolescents and emerging adults in Chinese.

RESPONSE: Thank you for your constructive comment. We have now expanded the manuscript at p.13-14 to deliver the information regarding practical implications.

REVISION: This study contains important practical implications that should be taken into account for mental health interventions among adolescents and emerging adults. First, educators or school-based psychologists should organize structured meetings with parents, either on-site or online, to introduce the concept of autonomy support and highlight the benefits of such interpersonal interaction styles for their children’s mental health. This introduction module is important for better conceptual understanding at the initial stage. During the following sections of these meetings, educators or school psychologists may provide specific guidance and strategies to parents regarding how to internalize these autonomy-oriented practices when interacting with their children. For instance, parents should welcome their children’s thoughts and attempt to incorporate their children’s perspectives when making decisions for them. Although parents should appropriately adjust their children’s irrational claims, they are recommended to use non-evaluative, non-controlling, flexible languages; parents should also provide possible solutions and present solid rationales when supervising their children on those decisions.

Second, educators or school psychologists should pay attention to the bridging role of self-esteem, particularly considering that, during adolescence and emerging adulthood, positive self-regard formation is crucial to student healthy development. Some school-based activities should be organized in this regard. For example, educators or school psychologists should let students to write down a list of moments when they feel well and a few things of which they feel proud. It is essential to reinforce these positive moments and experiences, stimulating positive perspectives on themselves.

Finally, during these initiatives, educators or school psychologists should regularly assess students’ perception of parental autonomy support and self-esteem. For those who consistently exhibit low parental autonomy support and self-esteem, educators or school psychologists may consider organizing individual psychological counseling sessions. In these scenarios, students and parents are encouraged to come together to talk with school psychologists. Professionals could better help them identify possible difficulties with these implementations and ultimately find ways to handle with these challenges that may affect how parents create autonomy-supportive styles and students feel about themselves.

Reviewer 2 Report

the topic is very interesting and the methodological structure of the study seems to me well thought out even if I would have explored more the role of the SES in the mediation of parental autonomy support

Author Response

1.The topic is very interesting and the methodological structure of the study seems to me well thought out even if I would have explored more the role of the SES in the mediation of parental autonomy support.

RESPONSE: Thank you for appreciating our work. We agree that family SES may influence the manifestation of parental autonomy support, which may further impact the direct and indirect correlations with individuals’ mental health under investigation. For this consideration, we regarded family SES as one of the covariates when testing our research hypotheses in the original manuscript. In responding to the reviewer’s suggestion, we elaborated more on the role of family SES when discussing our research findings at p. 12-13. Additionally, we highlight this in limitations and future directions of this study in a hope that much research will be conducted in this field to unfold (rather than control for) the role of family SES in the model tested. We have now added the following information on p.13. 

REVISION: Furthermore, family SES may influence the manifestation of parental autonomy support, which may further impact the direct and indirect correlations with individuals’ mental health under investigation. Family SES has shown to be positively related to parental autonomy support particularly in adolescents, as evidenced by the correlation analyses in Study 1. This is consistent with previous research indicating that SES can influence parenting practices (Liu & Wang, 2015; Waylen & Stewartbrown, 2010; Xu et al., 2019).  One possible explanation is that parents from higher SES families tend to use more positive practices, such as autonomy support, than those from lower SES families. Gaining psychological benefits from autonomy support, individuals are less likely to develop mental health difficulties. In sharp contrast, parents from low SES families often grapple with livelihood and encounter various life challenges due to restrained family economic situations (Waylen & Stewartbrown, 2010; Xu et al., 2019). Parents under this context may have limited time and energy in supervising their children using autonomy-supportive strategies because they may have difficulties with handling with their own psychological distress (Liu & Wang, 2015). Consequently, children raised under low SES families are more likely to suffer from mental health challenges due to a lack of autonomy support. In the current study, participants came from middle-class families, and thus the beneficial role of parental autonomy support in their mental health might be more pronounced (see p. 12-13).

Limitation: In this study, we statistically controlled for the levels of family SES by regressing these values on self-esteem and outcomes. Given that family SES may directly influence the manifestation of parental autonomy support, future studies should clarify the impact of family SES in the tested relationships by recruiting a more diverse sample with distinct socioeconomic characteristics (see p. 13).

REFERENCES:

Liu, L.; Wang, M. Parenting stress and children’s problem behavior in China: The mediating role of parental psychological aggression. J. Fam. Psychol. 2015, 29, 20–28. https ://doi.org/10.1037/fam00 00047

Waylen, A.; Stewartbrown, S. Factors influencing parenting in early childhood: a prospective longitudinal study focusing on change. Child Care Health Dev.2010, 36, 198–207. https ://doi.org/10.1111/j.1365-2214.2009. 01037.x

Xu, F.; Cui, W.; Xing, T.; Parkinson, M. Family socioeconomic status and adolescent depressive symptoms in a chinese low- and middle- income sample: the indirect effects of maternal care and adolescent sense of coherence. Front. Psychol. 2019, 10, 1–9. https ://doi.org/10.3389/fpsyg.2019.00819
